# Protective Human Anti-Poxvirus Monoclonal Antibodies Are Generated from Rare Memory B Cells Isolated by Multicolor Antigen Tetramers

**DOI:** 10.3390/vaccines10071084

**Published:** 2022-07-06

**Authors:** Xiuling Gu, Yufan Zhang, Wei Jiang, Dongfang Wang, Jiao Lu, Guanglei Gu, Chengfeng Qin, Min Fang

**Affiliations:** 1CAS Key Laboratory of Pathogenic Microbiology and Immunology, Institute of Microbiology, Chinese Academy of Sciences, Beijing 100101, China; guxl@im.ac.cn (X.G.); zhangyufan@im.ac.cn (Y.Z.); jiangw@im.ac.cn (W.J.); donwang@coh.org (D.W.); luj@im.ac.cn (J.L.); gugl@im.ac.cn (G.G.); 2University of Chinese Academy of Sciences, Beijing 101408, China; 3State Key Laboratory of Pathogen and Biosecurity, Beijing Institute of Microbiology and Epidemiology, Academy of Military Medical Sciences, Beijing 100071, China; 4International College, University of Chinese Academy of Sciences, Beijing 101408, China

**Keywords:** *Orthopoxvirus*, subunit vaccine, A33, tetramer, human antibody

## Abstract

Smallpox, an epidemic disease caused by *Orthopoxvirus variola*, was eradicated worldwide through immunization. The immunization against smallpox was discontinued in 1980. However, incidences of monkeypox virus infection in humans have occurred sporadically, and there is also great fear that engineered forms of poxvirus could be used as biological weapons. Therefore, monoclonal antibodies against poxvirus are urgently needed for the detection and treatment of poxvirus infection. The vaccinia virus’ extracellular envelope protein A33 is a potential candidate for a subunit vaccine. We used multi-fluorescence-labeled tetrameric A33 antigen to identify rare poxvirus-specific memory B cells from the PBMC of volunteers with vaccinia virus immunization more than 40 years ago. Despite extremely low frequencies of the poxvirus-specific memory B cells, we successfully sorted A33 tetramer-labeled single memory B cells and reconstructed the antibodies with the single-cell RT-PCR of the B-cell receptor. Among the monoclonal antibodies, one clone H2 exhibited high specificity and affinity with A33. H2 efficiently inhibited viral infection and spread in cells. Passive immunotherapy of H2 in mice protected mice from lethal infection when administered either prophylactically or therapeutically. These results suggest the potential of anti-A33 human-antibody-based detection and therapeutics for poxvirus infection.

## 1. Introduction

The *genus Orthopoxvirus* (OPV) is comprised of notable members such as variola virus (VARV), monkeypox virus (MPXV), vaccinia virus (VACV), cowpox virus (CPXV), and ectromelia virus (ECTV) [1]. Smallpox, caused by VARV, has periodically led to devastating epidemics throughout history and was eradicated following a worldwide vaccination campaign [2,3]. It has been more than four decades since the eradication of smallpox; vaccination programs have been halted and immunity has waned. Monkeypox is a zoonotic disease caused by MPXV [4]. In May 2022, multiple cases of MPXV infection were reported in the United States and several European countries. Furthermore, there were cases of some other zoonotic diseases including VACV in Brazil, CPXV in Europe, and buffalopox virus (BPVX) in India [5]. More seriously, the potential release of variola virus or engineered forms of poxvirus as an act of bioterrorism or the gradual transfer of zoonotic viruses from local hosts to humans emphasizes the importance of preparing for future poxvirus outbreaks.

VACV has been used as a live-attenuated vaccine to eradicate smallpox. Although VACV is an efficient vaccine, it routinely causes a pustular skin lesion, frequently induces lymphadenopathy and fever, and occasionally results in life-threatening disease [6]. Antibodies have been used as therapeutic drugs for more than 100 years. Monoclonal antibodies (mAbs) are effective biologics for use as therapeutic and diagnostic reagents [7]. Murine mAbs have a short half-life when used in humans and are relatively poor recruiters of effector function, such as antibody-dependent cellular cytotoxicity, and complement-dependent cytotoxicity [8]. Chimeric mAbs are constructed by coupling the rodent antigen-binding variable (V) domains to human constant (C) domains, are still 30% rodent in sequence and can elicit a significant anti-globulin response [4,9]. Humanized mAbs are created by transplanting the complementarity-determining regions (CDRs), which are the hypervariable loops involved in antigen binding from rodent antibodies into human V domains. However, the process to generate these molecules is arduous and has limitations [10]. With the development of new technology, it has become feasible to directly clone human antibody from a single B cell. Human mAbs have a longer circulation time in comparison with murine or chimeric antibodies [11].

There are two forms of infectious OPV: extracellular-enveloped virion (EV) and intracellular-mature virion (MV) [12]. MV particles are responsible for host-to-host spread, whereas EV particles are important for virus dissemination within the host as well as in cultured cells [6,13,14]. The reported target antigens recognized by neutralizing antibodies include A33 and B5 on the surface of EV, as well as L1, H3, A27, D8, A28, B7, and A17 of MV [15]. Among them, A33 plays an important role in the spread of the virus between cells. Immunization with A33 in mice elicits antibody responses and protects susceptible mice from lethal ECTV infection [16].

Over the past two decades, single B cell antibody technologies have become an effective approach to sample naive and antigen-experienced antibody repertoires generated in vivo [17]. Single B cell sorting and cloning of V_H_ and V_L_ is a powerful technology for generating neutralizing mAbs against infectious diseases, for example, mAbs against H1N1 or H5N1 influenza virus, SARS, and SARS-CoV-2 coronaviruses [18,19,20,21]. Normally, fluorescently labeled antigen has been used to identify B cells with particular BCR specificities by flow cytometry. However, the frequency of most specific memory B cells is low, and the signals generated by fluorescently labeled antigens are typically not bright and tend to overlap with unlabeled cell populations. Franz B et al. generated fluorescent antigen tetramers to isolate low-frequency memory B cells [22].

Here, we used multiple fluorescent A33 tetramers to identify and isolate rare memory B cells from the PBMC of a vaccinated volunteer.

## 2. Materials and Methods

### 2.1. Ethics Statement

This study for human peripheral blood was performed with the informed consent of the participant. The experimental design and protocols used for human peripheral blood in this study were approved by the Research Ethics Committee of the Institute of Microbiology, Chinese Academy of Sciences (permit number APIMCAS2019056). The animal protocols used in this study were approved by the Research Ethics Committee of the Institute of Microbiology, Chinese Academy of Sciences (permit number APIMCAS2017034).

### 2.2. Cells and Viruses

The 293T cells (ATCC, Manassas VA, USA. CRL-3216) and BSC-1 cells (ATCC, 3168) were cultured in complete DMEM medium. The 293-F cells (Invitrogen, Waltham, MA, USA. R790-07) were cultured in SMM 293-TI. All cells were cultured at 37 °C in 5% CO_2_.

Initial stocks of ECTV Moscow and VACV Western Reserve were obtained from Dr. Luis Sigal (Thomas Jefferson University) and amplified and quantified as described [23].

### 2.3. Production and Biotinylation of A33

The coding sequences for the extraviral domain of VACV Western Reserve A33 [24] were amplified by PCR from genomic DNA. For A33, the forward and reverse primers used were 5′AAACCATGGGCCATCACCATQCACCATCACTGCATGTCT-GCTAACGAGGCTG3′ and 5′AAAGGATCCGTTCATTGTTTTAACACAAAAATACTTTC3′, respectively. Avi-tag (GLNDIFEAQKIEWHE) [25] was fused to the C-terminus of A33 to mediate biotinylation. The expression, purification, and renaturation of A33 were performed as previously described [16]. Biotinylation of A33 protein was performed by using the biotin-protein ligase kit (GeneCopoeia^TM^, Rockville, MD, USA. B1001) in accordance with the instructions of the manufacturer.

The coding sequences for the extraviral domain of VARV A33 with N terminus 6×His tag and C-terminus Avi-tag were synthesized by GenScript (Nanjing, China) and directly cloned into the PET-28a(+). The purification and biotinylation of VARV A33 were the same as that of VACV A33.

### 2.4. Mice and Infection

Female C57BL/6 (B6) and BALB/c mice (6–8 weeks) were purchased from Vital River, China. For infection experiments, mice were transferred to a biosafety-level 2 room. B6 mice were infected in the left footpad with 30 μL of PBS containing 3000 plaque forming unit (PFU) of ECTV or infected intraperitoneal (i.p.) with 5 × 10^6^ PFU VACV.

Female BALB/c mice were infected i.p. with 5 × 10^6^ PFU VACV. H2 IgG was administered at 22 mg/kg into the indicated mice.

### 2.5. Production of Antisera

Anti-virus antisera were produced by infection B6 mice with ECTV or VACV. The sera were collected at five weeks pi and stored in aliquots at −80 °C.

### 2.6. Flow Cytometry and Tetramer Preparation

To detect A33-specific B cells in mice, single-cell suspensions were prepared from spleens obtained from naive B6 mice, B6 mice infected with ECTV, or B6 mice infected with VACV for 4–5 weeks. Red blood cells (RBC) were lysed with 0.84% NH_4_ Cl, the leukocytes were washed, and 2 × 10^6^ leukocytes were stained with the indicated surface antibodies at 4 °C for 30 min. Cells were analyzed with a BD LSRFortessa flow cytometer (BD Biosciences, San Jose, CA, USA), and the data were analyzed using FlowJo (Tree Star, Ashland, OR, USA).

All the tetramers were prepared freshly for each experiment. Biotinylated A33 (0.5 mg/mL) was incubated with APC-conjugated streptavidin (SA-APC, eBioscience, San Diego, CA, USA. 0.2 mg/mL), APC/Cy7-conjugated streptavidin (SA-APC/Cy7, eBioscience, 0.2 mg/mL), or Pacific Blue-conjugated streptavidin (SA-PB, eBioscience, 0.2 mg/mL) for 3 min at room temperature in a volume ratio of 2:1. The 2 μL fluorescent A33 tetramers were then used together with other antibodies for cell staining per sample.

The splenocytes of mice were stained as described previously [23]. To determinate the T cell responses, BALB/c mice were treated with H2 IgG or PBS before VACV infection. Seven days after the infection, mice were euthanized, and single-cell suspensions of each spleen were prepared in 5 mL complete RPMI medium. Following osmotic lysis of red blood cells with 0.84% NH_4_Cl, the spleen cells were washed, and 10^6^ cells were stimulated for 6 h at 37 °C with 2 × 10^5^ VACV infected or uninfected A20 cells in 96-well plates. Brefeldin A (BFA, Sigma, Saint Louis, MO, USA) was added after 4.5 h to block the secretory pathway and allow for the accumulation of cytokines inside the cells. The cells were then stained for cell-surface molecules, fixed, permeabilized, and stained for intracellular molecules using the Cytofix/Cytoperm kit (BD Biosciences) in accordance with the instructions of the manufacturer. Cells were analyzed with a BD LSRFortessa flow cytometer, and the data were analyzed using FlowJo.

Antibodies used for flow cytometry were anti-human CD19 (Biolegend, San Diego, CA, USA. 302208), anti-human IgM (Biolegend, 31450), anti-mouse/rat/human CD27 (Biolegend, 124214), anti-mouse CD8a (Biolegend, 53-6.7), anti-mouse CD4 (Sungene, Tianjin, China. M10043-02E), nti-mouse IFN-γ (BD, 554412), anti-mouse B220 (Biolegend, 103206), anti-mouse CD3 (Biolegend, 100312), and anti-mouse IgM (Biolegend, 406507).

### 2.7. Single-Cell Sorting

Blood samples were donated by a healthy volunteer. PBMC were isolated and stained with the indicated cell-surface markers. Single memory B cells were sorted into 96-well PCR plates as described previously [25]. Briefly, peripheral blood mononuclear cells (PBMC) were isolated from blood and then stained with the indicated cell-surface markers. Memory B cells were gated as CD19^+^ IgM^−^ CD27^+^. SA-APC A33^+^ and SA-PB A33^+^ double-positive memory B cells were sorted as single cells into 96-well PCR plates containing 20 μL/well of RT reaction buffer that included 5 μL of 5× First strand cDNA buffer, 0.5 μL of RNAseOut (Invitrogen, Carlsbad, CA, USA), 1.25 μL of DTT, 0.0625 μL of Igepal, and 13.25 μL of dH_2_O (Invitrogen, Carlsbad, CA, USA) The plates were briefly centrifuged and immediately stored at −80 ℃ until further processing. Cells were sorted on a BD FACS Aria IIIu, and the data were analyzed using FlowJo.

### 2.8. Single-Cell RT-PCR and Amplification of Antibody Variable-Region Sequences

The genes encoding IgG V_H_ and V_L_ chains were amplified by RT and nested PCR using the previously reported method [25]. All PCR products were purified and sequenced. Sequences were analyzed using the VBASE2 (http://www.vbase2.org) to identify variable-region gene segments.

### 2.9. Constructing and Expression of Single Chain Antibody (scFv)

The scFvs were constructed by connecting paired V_L_ and V_H_ with a (GGGS)_2_ linker, and an Ig leader sequence (METDTLLLWVLLLWVPGSTGD) was used as a signal peptide. A flag tag was fused to the C-terminus of scFv for easy detection. The constructs were cloned into pcDNA3.1 vector. The plasmids were transfected into 293 F cells, and the cell supernatants were collected after 48–72 h.

### 2.10. Production of H2 IgG

H2 IgG was generated by combining V_H_ and V_L_ with constant regions of IgG1 heavy and light chains, respectively. The constructed heavy- and light-chain plasmids were co-transfected into 293 F cells. H2 IgG was purified by protein A in accordance with the recommendations of the manufacturer.

### 2.11. ELISA

ELISA plates were coated with VACV A33, VARV A33 (50 μg/mL), ECTV (1 × 10^7^ PFU/mL), VACV (1 × 10^7^ PFU/mL), PR8 (1 × 10^7^ PFU/mL), or BSA (50 μg/mL) as indicated. ELISA was performed as described previously [16].

### 2.12. Comet-Inhibition Assay

BSC-1 cells were infected with ECTV (MOI = 0.01), and the indicated dilutions of antisera or antibodies were added. The assay was performed as described previously [16].

### 2.13. Surface-Plasmon Resonance Analysis

The affinity between H2 and A33 was measured at room temperature using a Biacore T100 system with CM5 chips (GE Healthcare, Chicago, IL, USA) in accordance with the recommendations of the manufacturer.

### 2.14. Histopathology

Histopathology was performed as described previously [26].

### 2.15. Statistics

Statistical analysis was performed using Prism software (GraphPad, San Diego, CA, USA). All statistical analyses were performed using an unpaired two-tailed Student’s *t*-test or two-way ANOVA test as applicable. When applicable, data were displayed as mean ± SEM. Unless indicated, all displayed data correspond to one representative experiment of at least three similar, independent experiments.

## 3. Results

### 3.1. Production and Biotinylation of VACV A33

Previous studies demonstrated that immunization of A33 protein protects susceptible mice from lethal mousepox [16]. Importantly, the extraviral region of VACV A33 and the orthology proteins of A33 from CPXV, VARV, MPXV, or ECTV are highly homologous (93.5%) (Figure 1A). The coding sequence of the extraviral domains of VACV A33 with a His-tag at the N terminus and an avi-tag at the C terminus was cloned into the expression vector pET-28a (+) (Figure 1B) for producing recombinant A33 in *E.coli*. The avi-tag mediates biotinylation by BirA enzyme. A33 was successfully expressed and purified (Figure 1C,D). Next, A33 was biotinylated and verified by Western blot (Figure 1E). Biotinylated A33 was recognized by both anti-ECTV and anti-VACV antisera (Figure 1F). These results suggest that biotinylation does not affect the binding of A33 to anti-OPV antibodies.

### 3.2. Tetramer-Based Screenings of A33-Specific Memory B Cells

To improve the efficacy and specificity of the screening for rare A33-specific memory B cells, we generated multiple color-labeled A33 tetramers. Biotinylated A33 was incubated separately with different fluorescent-conjugated streptavidin. The generated A33 tetramers were then used together with a panel of mAbs to identify the A33-specific memory B cells (Appendix A). To verify the feasibility of this strategy, we first infected B6 mice with 3000 pfu ECTV in the footpad. Four–five weeks post infection, leukocytes in the spleens were isolated for flow-cytometry analysis. As shown in Appendix A, we first gated on the CD3^−^B220^+^CD27^+^ B cells, and the A33-specific B cells were selected as the A33-Tet-PB^+^A33-Tet-APC^+^ double-positive cells. In the naive B6 mice, the proportion of A33-specific B cells was 0.53%. Surprisingly, the proportion of A33-specific B cells reached 39.7% in the ECTV-infected mice. The double staining of A33 tetramers defined a concentrated and clear cell population. This result was striking, and the importance of anti-A33 antibody responses during ECTV infection was emphasized. We further detected A33-specific memory B cells after VACV infection. B6 mice were infected i.p. with 5 × 10^6^ PFU VACV. Four–five weeks post infection, leukocytes in the spleens were isolated for flow cytometry analysis. We gated on B220^+^IgM^−^CD27^+^ class-switched memory B cells. While in the VACV-infected mice, A33-specific memory B cells comprised around 16% of the total memory B cells (Appendix A). Therefore, anti-A33 antibody responses account for a high proportion of the whole anti-poxvirus antibody responses.

Next, we used multiple-color A33 tetramers to isolate memory B cells from the PBMC of a healthy volunteer who was vaccinated against smallpox more than 40 years ago. As shown in Appendix A, A33-Tet-APC staining showed around 1% of A33-specific memory B cells among the total CD19^+^CD27^+^IgM^−^ memory B population, however, the staining was not very bright and contained some background staining. A33-Tet-PB staining showed a similar pattern. Whereas, A33-Tet-PB and A33-Tet-APC double staining defined a very rare cell proportion (0.045%) in the diagonal, which sharply reduced background staining. Single A33-Tet-PB^+^A33-Tet-APC^+^ double-positive memory B cells were sorted by flow cytometry and stored in 96-well plates for subsequent PCR amplification.

### 3.3. Amplification of Antibody Variable Region and Construction of Single-Chain Antibody

Single B cell mRNA amplification and nested PCR to amplify Ig V_H_ and V_L_ genes were performed as described previously [25]. From a total of 96 single cells, 19 pairs of heavy-chain and light-chain variable fragments were amplified, corresponding to an overall PCR efficiency of around 20% (Appendix A). We further analyzed the V, D, and J regions of 19 mAbs by the IMGT database. The results showed that all 19 antibody light chains were κ, and IGHV3-7*05 accounted for the highest proportion in the V region of IGH; IGHJ5*01 in the J region was the highest, and the proportions of IGKV1-5*03 and IGKJ2*01 were higher in IGK (Appendix A).

The CDRs of the V_H_ and V_L_ domains determine the antigen specificity of the Ig [27,28]. We then compared the sequences of the amplified 19 pairs of V_H_ and V_L_ (Appendix A). A7, A8, A10 and A11 had identical V_H_ and V_L_, indicating that the cells might come from one original B cell clone. B9 and H2 also had identical V_H_ and V_L_. In addition, nine clones (B1, B2, B7, B11, C1, C3, C4, C6, and D7) differed from A7 in 1–8 amino acids in the framework and CDR regions. The remaining B3, C2, and C8 demonstrated a larger number of amino acid differences, indicating distant clonal relationship. Thus, based on the difference in sequences, especially within CDRs, we selected nine pairs of V_H_ and V_L_ to construct a single-chain antibody (scFv) (Appendix A) for rapid and easy screening of the antigen specificity of the Ig.

We transfected the nine scFv plasmids into 293F cells, respectively. Among the nine constructs, only H2, A7, B3, and B6 were expressed (Appendix A). Next, ELISA plates were coated with A33 protein and tested for recognition by the scFvs. As shown in Figure 2A (left panel), only H2 and A7 bound to A33, but the affinity of A7 was lower than that of H2. Then, 293F cell culture supernatant was used as a negative control. As H2 showed higher binding capacity to A33, we next determined the binding specificity of H2 scFv. ELISA plates were coated with A33, influenza virus A/Puerto Rico/8/34 (PR8), or vesicular stomatitis virus (VSV). H2 is only bound to A33, not PR8 or VSV (Figure 2A, right panel). A33 is highly conserved between OPVs. As expected, H2 scFv bound both ECTV and VACV (Figure 2B).

### 3.4. H2 scFv Blocks Virus Replication in Cell Culture

We used surface-plasmon resonance (SPR) to determine the affinity of H2 scFv to A33. As shown in Figure 2C, the binding affinity (KD) between H2 scFv and A33 was 59.4 nM. Our previous studies demonstrated that antisera against A33 inhibit comet formation in ECTV-infected cells [16]. Therefore, we determined whether comet formation by ECTV could be inhibited by H2 scFv. As shown in Figure 2D, H2 scFv inhibited comet formation as well as viral infection. The anti-ECTV antisera were used as a positive control. Anti-ECTV antisera or H2 scFv both resulted in a marked reduction in plaques in addition to the inhibition of comet formation. It may be because the antisera or scFv block released virions to infect adjacent cells, thus preventing the formation of plaques. Previous studies have shown that Abs against the EV B5 protein are mainly responsible for the EV-neutralizing capacity of VACV, as measured by a plaque-reduction assay [29]. Our experiments demonstrated that Ab against EV A33 also can reduce plaque formation. Collectively, H2 scFV shows functional activity in vitro.

### 3.5. Purification and Functional Characteristics of H2 IgG

Next, we generated H2 IgG by combining V_H_ and V_L_ with constant regions of IgG1 heavy and light chains, respectively. The purified H2 IgG was verified by SDS-PAGE (Figure 3A). H2 IgG specifically recognized the A33 protein, ECTV, and VACV, but not PR8 or VSV, which was consistent with the results of scFv (Figure 3B,C). In addition, H2 IgG inhibited comet formation and viral replication in ECTV-infected BSC-1 cells (Figure 3D). Next, we wanted to verify whether H2 IgG can bind to the orthology proteins of A33 from VARV or MPXV. The coding sequences of those proteins were synthesized. The orthology proteins of A33 from VARV (Figure 3E,F) or MPXV (Figure 3H,I) were expressed and purified. As shown in Figure 3G,J, H2 IgG recognized the orthology proteins of A33 from VARV and MPXV. This result suggests that H2 IgG may inhibit the spread of the VARV and MPXV as well and is a broad protective mAb against OPV.

### 3.6. H2 IgG Protects Mice against VACV Infection

To test whether H2 IgG can protect mice from OPV infection, BALB/c mice were infected i.p. with 5 × 10^6^ PFU VACV. H2 IgG or PBS was administered 4 h before and 48 h after infection (Figure 4A). As shown in Figure 4B, both groups of mice lost weight after infection, and the weight loss in the H2 IgG group remained relatively stable from 1 dpi to 5 dpi, with mice starting to recover the weight from day 6. However, the mice of the PBS group continued to lose weight from 1 dpi to 10 dpi. Only around 20% of the mice in the PBS-treated control group survived after infection, while the mice injected with H2 IgG had a survival rate of up to 80% (Figure 4C). Thus, H2 IgG significantly protected mice from VACV infection when administered prophylactically.

Next, we tested whether H2 IgG can protect mice after VACV infection. BALB/c mice were infected with VACV. After 48 h, the mice were injected with H2 IgG or PBS (Figure 4D). Again, both groups of mice experienced weight loss after infection, however, the H2 IgG group started to regain the weight from 5 dpi (Figure 4E). Again, H2 IgG administration after VACV infection significantly protected mice from death (Figure 4F). Collectively, these results demonstrated that H2 IgG can protect mice from VACV infection both prophylactically and therapeutically.

### 3.7. H2 IgG Promotes Anti-VACV T Cell and Ab Responses

T cell responses are known to be important for protection against OPV infections [23,30,31]. Therefore, we next determined whether administration with H2 IgG has any effects on the T cell responses. Mice were first administrated with H2 IgG, or PBS, then challenged with VACV, and 7 days later their spleens were collected. Compared to naïve mice, the spleens of the mice in the PBS group were reduced, while the spleens of the mice in the H2 IgG group were significantly enlarged. The total number of splenocytes in the naive mice was 3.9 ± 0.15 × 10^7^, in the PBS group it was 2.6 ± 0.35 × 10^7^, and in the H2 IgG group it was 8.3 ± 0.29 × 10^7^, which increased significantly (Figure 5A). Moreover, after VACV infection, the percentage of CD4^+^ T cells decreased. However, the CD4^+^ T cells in the H2 IgG group increased, while those in the PBS group decreased. In addition, the percentage of virus-specific CD4^+^ T cells increased in the H2 IgG group compared with that of the PBS group, as determined by intracellular IFN-γ staining following in vitro restimulation with VACV-infected cells (Figure 5B).

The percentages of CD8^+^ T cells remained relatively stable after VACV infection, while the cell number of CD8^+^ T cells in the H2 IgG group increased. Markedly, the percentage of IFN-γ^+^ CD8^+^ T cells increased in the H2 IgG group, reaching around 30%, which was significantly higher than that of the PBS group (Figure 5C). Therefore, administration of H2 IgG prevented lymphocyte depopulation and promoted stronger anti-VACV CD4^+^ and CD8^+^ T cell responses.

Next, we determined the viral loads in the spleen and liver. As shown in Figure 5D, H2 IgG administration resulted in significantly decreased viral titers in both the spleen and liver. Moreover, even though both groups had necrotic foci in their livers, those of the H2 IgG group were more heavily infiltrated with lymphocytes than those of the PBS group (Figure 5E). In addition, at 1 week pi, anti-VACV IgM was detected in both infected groups, while almost no anti-VACV IgG was detected as we only detected murine IgG. However, the IgG level in the H2 IgG group was substantially higher than that of the PBS group at 3 weeks pi (Figure 5F). Therefore, H2 IgG promoted anti-VACV Ab responses as well. Together, these results demonstrated that administration of H2 IgG resulted in stronger T cell and Ab responses.

## 4. Discussion

Previous studies have suggested that antibodies are sufficient to protect against OPV infections in mice and monkeys [32,33]. A recent study showed that poxvirus proteome microarrays can be valuable for screening and monitoring smallpox-vaccine-induced humoral immune responses [34]. Vaccinia immune globulin (VIG) is manufactured from vaccinia vaccine-boosted plasma; however, this production method is not ideal because of its limited availability and the risk of contamination with blood-borne infectious agents [35,36]. Abs in VIG recognize many antigen targets, including the surface proteins of both the MV (A27, L1, H3, D8, A28, A13, and A17) and EV (B5 and A33) virion forms of VACV [37]. Protective mAbs had been reported against A33, B5, L1, and H3, such as anti-B5 chimpanzee/human mAbs 8AH8AL and 8AH7AL [38] and anti-L1 mouse mAb 7D11 [39]. Human mAbs against H3 and B5 were produced by using KM mice [40,41]. A large panel of OPV-specific human Abs was generated by hybridoma cell lines, and the mAb panel contained Abs for at least 12 antigens: D8, B5, A33, H3, L1, A27, I1, A25, F9, A28, A21, and H5 [42]. Recently, two human scFvs (SC34 and SC212) that bound to VACV were isolated by a phage library constructed from the B cells of VACV vaccine-boosted volunteers [35]. Furthermore, it is feasible to use mRNA constructs to produce multiple mAbs against poxvirus in vivo [43].

EV-membrane protein A33 plays an important role in effective cell to cell spread within the host [44,45]. A33 is also required for the proper formation of infectious EEV [45,46]. Anti-A33 mAbs can be protective in vivo [15,16,47]. Anti-A33 chimpanzee/human mAbs 6C, 12C, and 12F exhibited higher protective efficacy than a mouse anti-A33 mAb or a human VIG [15]. Furthermore, 6C had been humanized [48]. Importantly, A33 is highly homology between CPXV, VARV, VACV, MPXV, and ECTV, which indicates that anti-A33 mAbs can be used for evaluating CPXV and VACV vaccination or MPXV and ECTV infection. Here, we sorted A33-specific memory B cells by using multicolor tetramer staining and obtained one mAb H2 with a high specificity and affinity to A33.

The mAb H2 showed functional activity in vitro. Moreover, mAb H2 protected BALB/c mice from lethal VACV infection either administered prophylactically or therapeutically. T cell responses are known to be important in protection against OPV infections [49]. We found that the number of CD4^+^ and CD8^+^ T cells and the T cell responses all increased in the mice administered with mAb H2. Further, administration of H2 also stimulated stronger Ab responses in the mice. Although H2 is protective against VACV infection, it does not provide complete protection. Random mutation of the CDR region or prediction of the affinity of mAbs to A33 by using evolutionary information from the family of related sequences [50] may be feasible strategies to improve the protective effect of H2. In addition, the combination of H2 with other mAbs, such as mAbs aginast B5 and MV antigens (such as L1, H3, B7), might be able to provide full protection against poxvirus infection.

Another important observation of our study is the direct visualization of the anti-A33 Ab responses during OPV infections. The proportion of A33-specific B cells reached 39.7% in the ECTV-infected mice. During VACV infection, A33-specific memory B cells comprised around 16% of total memory B cells. A previous study generated a large panel of OPV-specific human mAbs and found that most of the neutralizing mAbs recognized one of six proteins from OPV, D8, L1, B5 A33, A27, or H3 [42]. Still, the large proportion of anti-A33-specific B cells were surprising, especially during ECTV infection. However, the mice were housed in SPF condition and infected only once with the indicated OPV, and we detected the memory B cells at 4–5 weeks post infection, so the proportion may decline with time. Further, it may be different with human beings as they are consistently challenged with different pathogens. A caveat of our data is that another unrelated virus infection to compare and evaluate the proportion of antigen-specific B cells is lacking. We will explore other virus-specific B cells in our future investigations. Nonetheless, our data still suggested that anti-A33 antibody responses account for a high proportion of the whole anti-poxvirus antibody responses. As ECTV is a natural mouse pathogen, which spreads massively in mice, the high proportion of anti-A33 Ab responses highlights the importance of anti-A33 Abs in protection. Since VARV specifically infects humans, and human beings are the only natural host for VARV, anti-A33 Ab responses might also play an important role in curbing virus spread in vivo. Our H2 mAb, which also binds to the orthology proteins of A33 from VARV and MPXV, might be used for the treatment of complications of smallpox vaccination and MPXV infection. Therefore, H2 is a broad protective mAb against OPV, which provides a new approach for the prevention and treatment of OPV infections.

## Figures and Tables

**Figure 1 vaccines-10-01084-f001:**
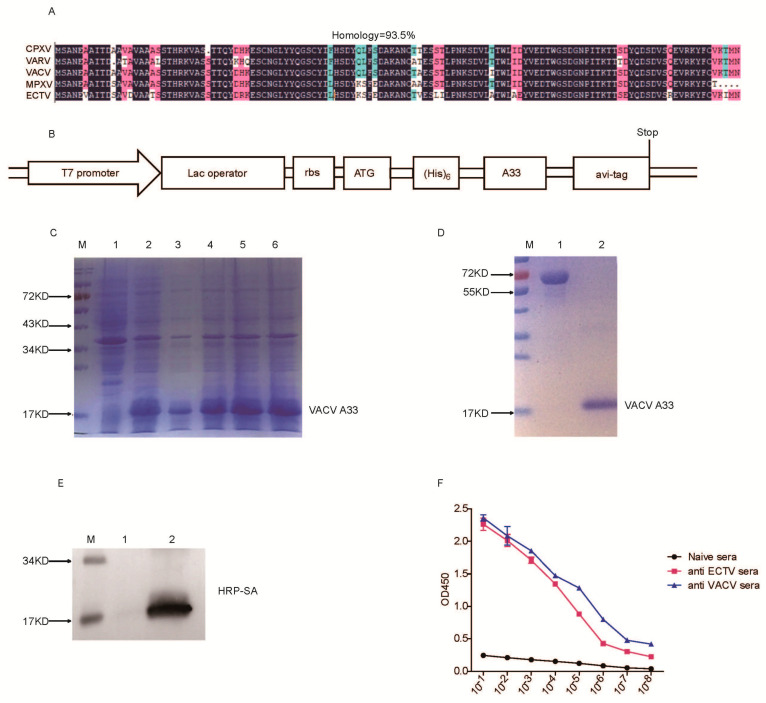
Production and biotinylation of A33. (**A**) DNAMAN software comparison of the homology of amino acid sequences of A33 extracellular region from VARV, VACV, MPXV, and ECTV. (**B**) Scheme of the expression vector. (**C**) Expression of A33 in BL21(DE3) cells induced with IPTG. Lane 1: control BL21(DE3) cells (vector). Lane 2, 3, 4, 5, 6: different clones of BL21(DE3) cells transfected with pET/A33. (**D**) Purification of A33. Lane 1: 1 mg/ mL BSA for comparison; Lane 2: purified and refolded A33 protein. (**E**) Western blot detection of A33 biotinylation. Lane 1: A33 protein; Lane 2: biotinylated A33 protein. (**F**) Biotinylated A33 was recognized by anti-ECTV sera and anti-VACV sera. Sera from naive mice were used as a negative control.

**Figure 2 vaccines-10-01084-f002:**
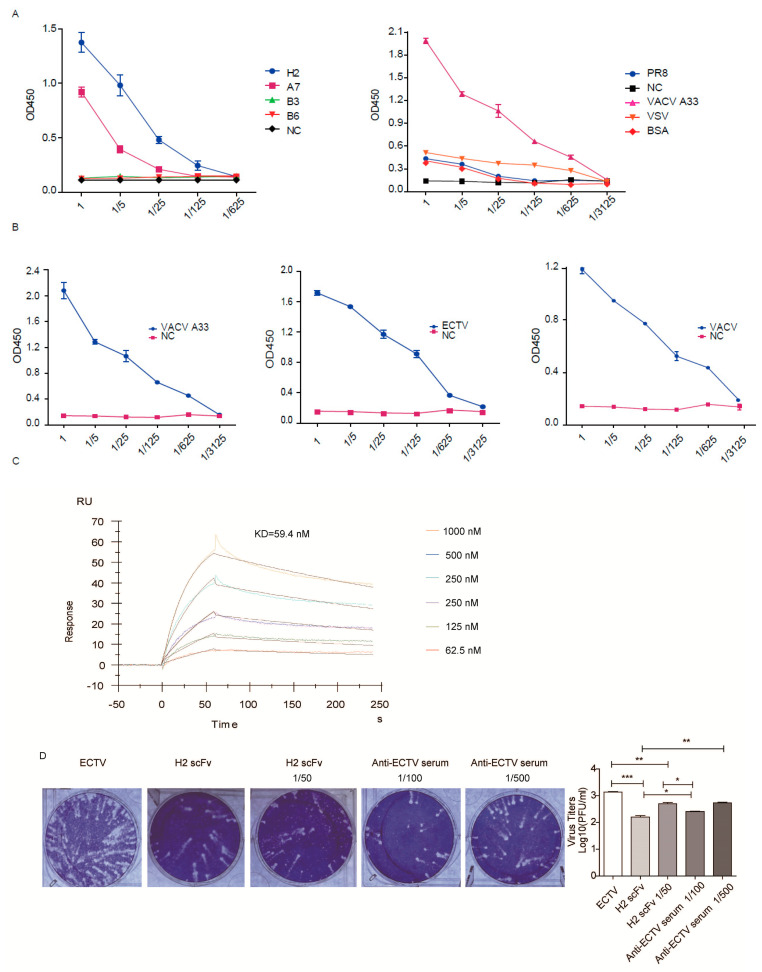
Functional identification of ScFvs. (**A**) ELISA plates were coated with A33 and then incubated with the indicated scFvs (left panel); ELISA plates were coated with A33, PR8, VSV, or BSA and then incubated with H2 scFv (right panel). (**B**) ELISA plates were coated with A33, ECTV, or VACV, and then incubated with serial dilutions of H2 scFv. Uncoated wells were used as a negative control (NC). (**C**) Representative SPR sensorgrams of H2 scFv binding to A33. (**D**) Comet-inhibition assay. * *p* < 0.05, ** *p* < 0.01, *** *p* < 0.001.

**Figure 3 vaccines-10-01084-f003:**
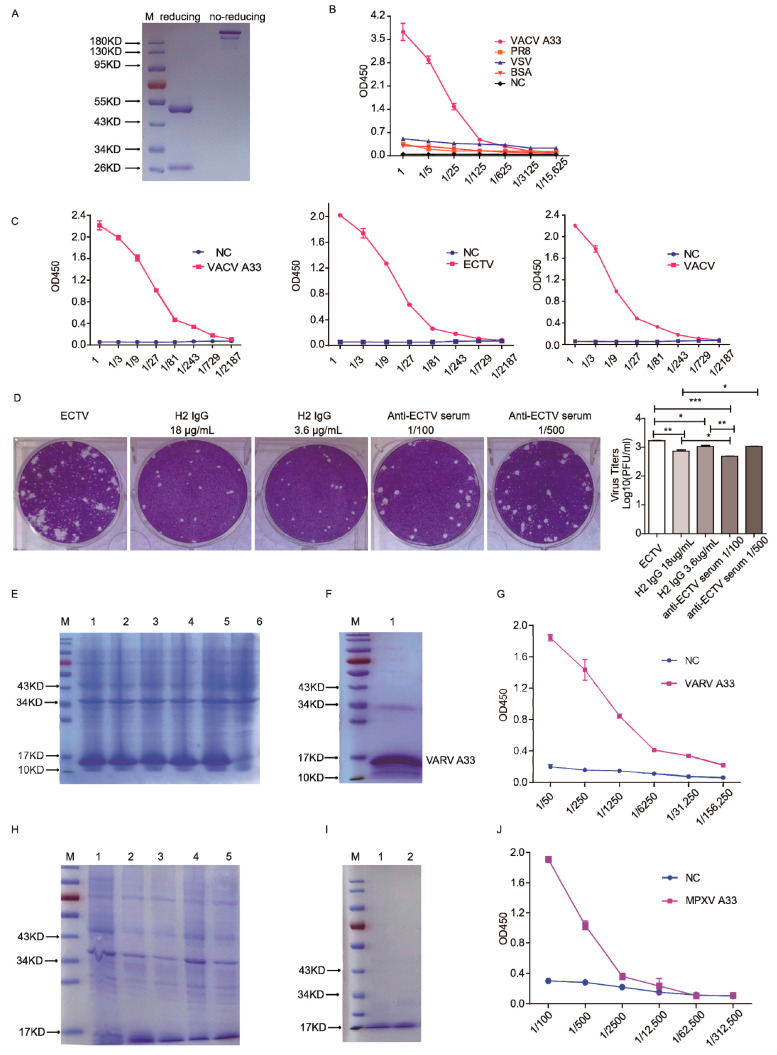
Purification and functional identification of H2 IgG. (**A**) SDS-PAGE of the purified H2 IgG under reducing (left) and non-reducing conditions (right). (**B**) The wells of ELISA plates were respectively coated with A33, VSV, PR8, or BSA and then incubated with serial dilutions of H2 IgG. (**C**) The wells of ELISA plates were coated with A33, ECTV, or VACV, respectively, and then incubated with serial dilutions of H2 IgG. (**D**) Comet-inhibition assay. * *p* < 0.05, ** *p* < 0.01, *** *p* < 0.001. (**E**) Expression of VARV A33 in BL21 (DE3) cells induced with IPTG. Lane 1, 2, 3, 4, 5: different clones of BL21 (DE3) cells transfected with pET/VARV A33. Lane 6: control BL21 (DE3) cells (vector). (**F**) Purification of VARV A33. Lane 1: purified and refolded VARV A33 protein. (**G**) The wells of ELISA plates were coated with VARV A33 and then incubated with serial dilutions of H2 IgG. (**H**) Expression of MPXV A33 in BL21 (DE3) cells induced with IPTG. Lane 1: control BL21 (DE3) cells (vector). Lane 2, 3, 4, 5: different clones of BL21 (DE3) cells transfected with pET/MPXV A33. (**I**) Purification of MPXV A33. Lanes 1 and 2: purified and refolded MPXV A33 protein. (**J**) The wells of ELISA plates were coated with MPXV A33 and then incubated with serial dilutions of H2 IgG.

**Figure 4 vaccines-10-01084-f004:**
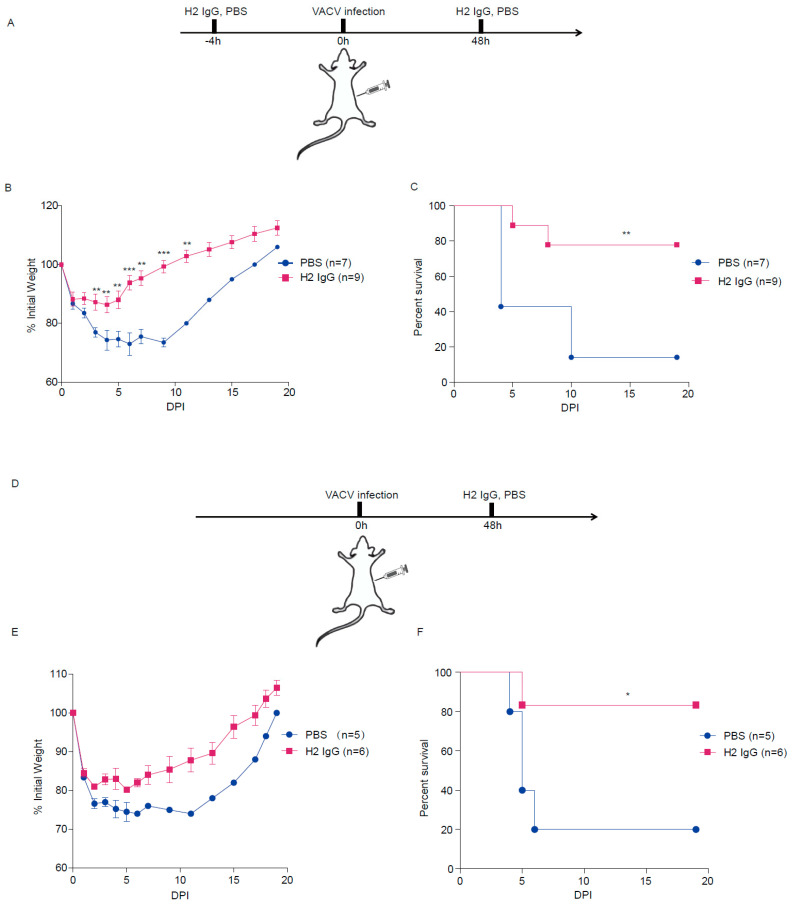
Prophylactic and therapeutic protection in mice by H2 IgG. Female BALB/c mice were infected i.p. with 5 × 10^6^ PFU VACV. H2 IgG was administered in mice (22 mg/kg) at the indicated time points. (**A**) Schematic diagram of prophylactic treatment with H2 IgG in mice. (**B**) Weight change following VACV infection. (**C**) Survival curves following VACV infection. (**D**) Schematic diagram of treatment with H2 IgG in mice. (**E**) Weight change following VACV infection. (**F**) Survival curves following VACV infection. Data are from at least two independent experiments, with 3–5 mice per group in each experiment. * *p* < 0.05, ** *p* < 0.01, *** *p* < 0.001.

**Figure 5 vaccines-10-01084-f005:**
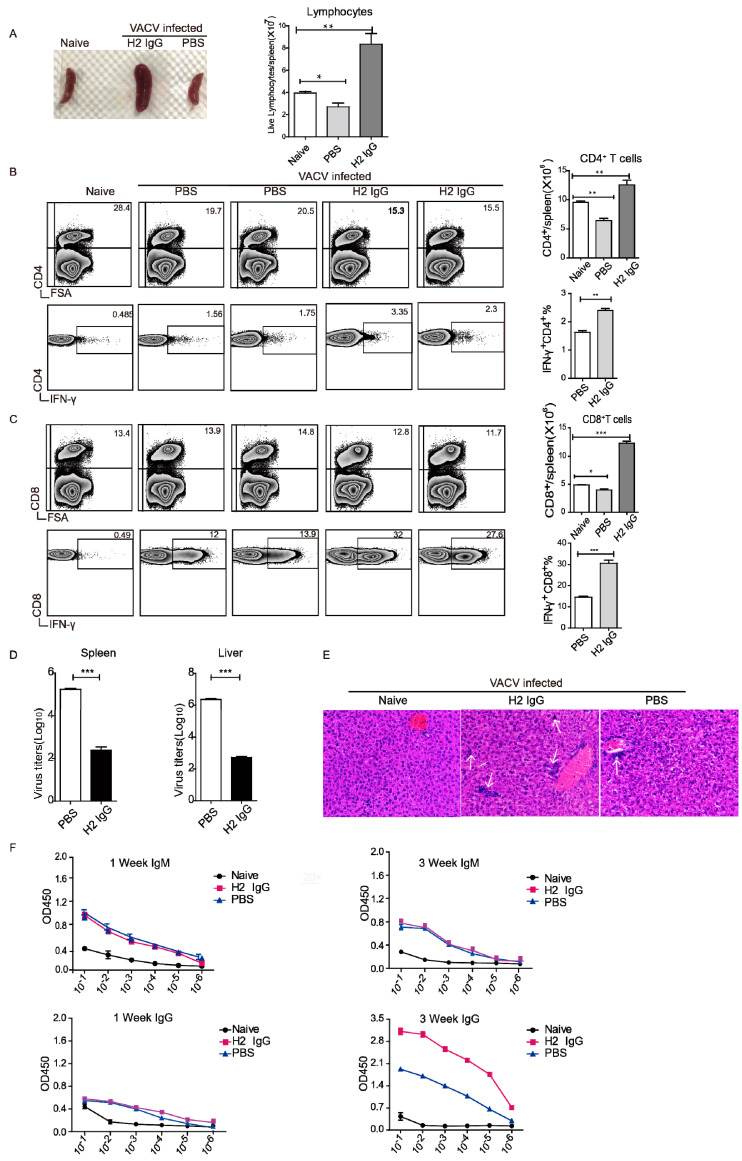
H2 IgG promotes anti-VACV T cell and Ab responses. (**A**) Indicated mice were administered with H2 IgG or PBS before VACV infection. The mice were euthanized at 7 dpi and the size of the spleens was observed (left panel). Total lymphocytes per spleen were counted (right panel). (**B**) Left panel: flow-cytometry analysis showing the proportion of CD4^+^ T cells in the spleen and the proportion of IFN-γ^+^ cells among CD4^+^ T cells. FSA: forward scatter amplitude. Stained as indicated. Right panel: total CD4^+^ T cells per spleen and proportion of IFN-γ^+^ cells among CD4^+^ T cells. (**C**) Left panel: flow-cytometry analysis showing the proportion of CD8^+^ T cells in the spleen and the proportion of IFN-γ^+^ cells among CD8^+^ T cells. Stained as indicated. Right panel: total CD8^+^ T cells per spleen and proportion of IFN-γ^+^ cells among CD8^+^ T cells. Numbers indicate the proportion in the nearest quadrant. (**D**) Virus titers in spleens and livers were determined at 7 dpi. (**E**) H&E stains of the liver (original magnification, ×20) at 7 dpi. The arrow pointed to the lymphocyte-infiltration area. (**F**) anti-VACV IgM or IgG Ab responses were determined at 1 week and 3 weeks pi. Data are from at least two independent experiments, with 3 mice per group in each experiment. * *p* < 0.05, ** *p* < 0.01, *** *p* < 0.001.

## Data Availability

Not applicable.

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
