# Peer review of "Protective Human Anti-Poxvirus Monoclonal Antibodies Are Generated from Rare Memory B Cells Isolated by Multicolor Antigen Tetramers"

_vaccines, 2022, doi:10.3390/vaccines10071084_

Round 1

Reviewer 1 Report

The team has successfully sorted A33 tetramer-labeled single memory B cells and reconstructed an24 antibodies with single-cell RT-PCR of the B cell receptor. The whole study is well-designed and the overall presentation is clear.

I understand that extraviral region of VARV, VACV, MPXV, and ECTV is highly homology (92.68%), and the elevated anti-A33 antibody responses have been also observed for both VACV and ECTV infection. However, further discussions on the application of H2 mAb in VARV and MPXV vaccination are not strongly persuasive (line 397-401). 

How about CPVX? Lower sequence homology? Any chance of this H2 mAb to be applied in CPVX vaccination?

Would expect more extensive and deep discussions on potential feasible strategies which could improve the current performance of H2 mAb.

Citation of more recent literatures could help support the significance of this study in a better way.

Author Response

Point by point response to reviewers

We would first like to thank the reviewers for their helpful comments. Reviewers’ questions are in bold and our reply in regular type.

Comments and Suggestions for Authors

The team has successfully sorted A33 tetramer-labeled single memory B cells and reconstructed an24 antibodies with single-cell RT-PCR of the B cell receptor. The whole study is well-designed and the overall presentation is clear.

We thank the reviewer for the positive comments.

I understand that extraviral region of VARV, VACV, MPXV, and ECTV is highly homology (92.68%), and the elevated anti-A33 antibody responses have been also observed for both VACV and ECTV infection. However, further discussions on the application of H2 mAb in VARV and MPXV vaccination are not strongly persuasive (line 397-401). 

We thank the reviewer for this suggestion. During the submission of the manuscript, we synthetized and expressed extraviral region of A33 from MPXV, and H2 mAb recognized A33 from MPXV as well. The data were included in Fig.3H-3J in the revised manuscript. Thus, H2 mAb may inhibit the spread of VARV and MPXV, we extended discussion on the revised manuscript as suggested by the reviewer.

How about CPVX? Lower sequence homology? Any chance of this H2 mAb to be applied in CPVX vaccination?

We compared the sequence homology of CPVX as suggested by the reviewer. There is high homology between CPVX, VARV, VACV, MPXV, and ECTV (93.5%). We replaced the Fig.1A with the new comparison in the revised manuscript and discussed the possibility of H2 mAb to be applied in CPXV vaccination.

Would expect more extensive and deep discussions on potential feasible strategies which could improve the current performance of H2 mAb.

We thank the reviewer for the question, we discussed potential feasible strategies to improve the current performance of H2 mAb in the revised manuscript.

Citation of more recent literatures could help support the significance of this study in a better way.

We added several recent literatures in the revised manuscript as the reviewer suggested. 

Reviewer 2 Report

In this manuscript titled “Protective human anti-poxvirus monoclonal antibodies are generated from rare memory B cells isolated by multicolor antigen tetramers”, the authors identified an A33-specific anti-poxvirus monoclonal antibody from a vaccinated donor through a single cell sorting and sequencing protocol. And the identified antibody clone H2 was able to treat VACV infection in a mouse model and induced a stronger T cells response compared to PBS treatment. Overall, this is a timely study and the manuscript is well written for most parts. However, due to the lack of proper controls and of detailed data, I am not convinced that the H2 monoclonal antibody was truly protective. Therefore, a major revision is needed.

Major issues:

1. The process of how these A33-specific single B cells were sorted was not stated with sufficient details, such as what is the concentration of A33-biotin and step-fluorescence used for staining of these A33-specific cells. Without such details, it is impossible to reproduce such study.

2. The authors didn’t show the sequence of these sorted out antibodies, which makes it impossible to reproduce or validate this study.

3. Figure 2D and 3D, these virus neutralization results are not convincing due to lack of repeats and quantification. The authors probably want to quantify these results.

4. Line 276, I believe the authors were referring to figure 3E and 3F, instead of 5E and 5F

5. In figure 4, I don’t understand how the mouse weight could be compared between PBS treated and H2 IgG treated mice. Based on the survival data, a large proportion of PBS treated mice died early. How did the authors compared the weight if most of the mice died?

6. In the mouse model (figure 4 and 5), PBS treatment is not a proper control to demonstrate H2 IgG is protective. H2 IgG is human origin, so it has the potential to generate immune responses in mouse which are actually against human IgG rather than against A33. Isotype control antibody treatment is needed to show that the differences seen in figure 4 and 5 are protective immune responses generated by A33-specific H2 IgG.

7. Line 391-393, “The proportion of A33-specific B cells reached 39.7% in the ECTV infected mice. During VACV infection, A33 specific memory B cells comprised around 16% of total memory B cells”, this statement is quite shocking because virus infections typically won’t generate such a large percentage of virus specific antibody. Again, as mentioned in major issue #1, since the authors didn’t describe the details of the staining and sorting procedure, especially the concentration of the reagents used, it is impossible to validate and repeat this finding. Also, the authors didn’t include proper controls, for example, did other related or unrelated viruses generate such a large proportion of virus specific B cells too? Did the authors sort out these virus specific B cells to look at their virus neutralization ability?

Author Response

Point by point response to reviewers

We would first like to thank the reviewers for their helpful comments. Reviewers’ questions are in bold and our reply in regular type.

 Comments and Suggestions for Authors

In this manuscript titled “Protective human anti-poxvirus monoclonal antibodies are generated from rare memory B cells isolated by multicolor antigen tetramers”, the authors identified an A33-specific anti-poxvirus monoclonal antibody from a vaccinated donor through a single cell sorting and sequencing protocol. And the identified antibody clone H2 was able to treat VACV infection in a mouse model and induced a stronger T cells response compared to PBS treatment. Overall, this is a timely study and the manuscript is well written for most parts. However, due to the lack of proper controls and of detailed data, I am not convinced that the H2 monoclonal antibody was truly protective. Therefore, a major revision is needed.

 We thank the reviewer for the insightful comments.

Major issues:

  1. The process of how these A33-specific single B cells were sorted was not stated with sufficient details, such as what is the concentration of A33-biotin and step-fluorescence used for staining of these A33-specific cells. Without such details, it is impossible to reproduce such study.

We thank the reviewer for the question. During the initial submission of the manuscript, we simplified the Material and Methods (MM) part to make the manuscript more concise. We added the details of the single B cell sorting in the MM part in the revised manuscript, and also included extended MM as supplemental data to make the procedures of other experiments clearer.  

  1. The authors didn’t show the sequence of these sorted out antibodies, which makes it impossible to reproduce or validate this study.

We agree with the reviewer that the sequences of these sorted out antibodies are important for other researchers to reproduce or validate our study. Thus, we included the VH and VL sequences of H2 and A7 as supplemental data in the revised manuscript.

  1. Figure 2D and 3D, these virus neutralization results are not convincing due to lack of repeats and quantification. The authors probably want to quantify these results.

We are sorry that we did not state the experiments clearly. All the experiments in the manuscript were repeated at least three times unless indicated otherwise. All the displayed data correspond to one representative experiment of at least three similar independent experiments. We added the information in the Statistics of the MM part the revised manuscript. We also added quantification for Fig. 2D and 3D in the revised manuscript as suggested by the reviewer.

  1. Line 276, I believe the authors were referring to figure 3E and 3F, instead of 5E and 5F

We thank the reviewer for careful reading. We corrected the mistakes in the revised manuscript.

  1. In figure 4, I don’t understand how the mouse weight could be compared between PBS treated and H2 IgG treated mice. Based on the survival data, a large proportion of PBS treated mice died early. How did the authors compared the weight if most of the mice died?

We weighted the mice daily during the experiments. At each time points, the weights corresponded to the average weight of all the surviving mice in each group at that time. Because a large proportion of PBS treated mice died early, the weights were the average from a few surviving mice at later time points post infection. That is also the reason for the rapid recovery of body weights after 10 DPI in the PBS treated group as only a few mice survived and recovered from the infection.

  1. In the mouse model (figure 4 and 5), PBS treatment is not a proper control to demonstrate H2 IgG is protective. H2 IgG is human origin, so it has the potential to generate immune responses in mouse which are actually against human IgG rather than against A33. Isotype control antibody treatment is needed to show that the differences seen in figure 4 and 5 are protective immune responses generated by A33-specific H2 IgG.

We thank the reviewer for this question. We totally agree with the reviewer that human IgG isotype control antibody treatment is ideal for the control experiments. In fact, we tried to purchase human IgG isotype control antibody before doing those experiments. However, because of the COVID-19 pandemic, many reagents are unavailable or will take very long time to obtain. At that time, there was no available human IgG isotype control antibody, thus we had to use PBS as a control. H2 IgG is human origin, the mice might generate immune responses against H2 IgG. However, as the primary humoral responses against H2 IgG will take longer time (normally one-two weeks after H2 IgG administration), the protective effect of H2 IgG should take place at early time (within one week) as the mice died early after VACV infection, which is earlier than the mice would generate immune responses against H2 IgG. Thus, even PBS was not an ideal control, we think that the results were still meaningful.

  1. Line 391-393, “The proportion of A33-specific B cells reached 39.7% in the ECTV infected mice. During VACV infection, A33 specific memory B cells comprised around 16% of total memory B cells”, this statement is quite shocking because virus infections typically won’t generate such a large percentage of virus specific antibody. Again, as mentioned in major issue #1, since the authors didn’t describe the details of the staining and sorting procedure, especially the concentration of the reagents used, it is impossible to validate and repeat this finding. Also, the authors didn’t include proper controls, for example, did other related or unrelated viruses generate such a large proportion of virus specific B cells too? Did the authors sort out these virus specific B cells to look at their virus neutralization ability?

We thank the reviewer for the question. We included details for the staining and sorting procedure in the revised manuscript. A large panel of OPV-specific human mAbs were generated in a previous study, and the authors found that most of the neutralizing mAbs recognized one of six proteins from OPV, D8, L1, B5 A33, A27, or H3 (http://dx.doi.org/10.1016/j.cell.2016.09.049). Still, we were also surprised by the large proportion of anti-A33 specific B cells, especially during ECTV infection. However, as the mice were housed in SPF condition and infected only once with the indicated OPV and we detected the memory B cells at 4-5 weeks post infection, the proportion might decline with time. Further, it may different with human beings as they are consistently challenged with different pathogens. We agree with the reviewer that other unrelated virus infection for comparison of the proportion of antigen-specific B cells would be helpful. We think the tetrameric antigen is a useful tool to directly visualize the antigen-specific B cells, we will explore other virus specific B cells in our future investigations. 

Round 2

Reviewer 2 Report

Regarding issue #7, the authors had clearly stated the caveats on the high percentage of "A33-specific" B cells in the response. Then, the authors should clearly state these caveats or possible explanation in the discussion too.

All other issues resolved.

Author Response

Point by point response to reviewers

We would first like to thank the reviewer for the helpful comments. Reviewers’ questions are in bold and our reply in regular type.

Comments and Suggestions for Authors

 Reviewer 2:

 Regarding issue #7, the authors had clearly stated the caveats on the high percentage of "A33-specific" B cells in the response. Then, the authors should clearly state these caveats or possible explanation in the discussion too.

We thank the reviewer for the suggestion. We added discussion about the caveats or possible explanation of the high percentage of A33-specific B cells in the revised manuscript.

All other issues resolved.

We thank the reviewer for the supportive comments on our revision.